# DapperFL: Domain Adaptive Federated Learning with Model Fusion Pruning for Edge Devices

**Yongzhe Jia**[1], **Xuyun Zhang**[2], **Hongsheng Hu**[3], **Kim-Kwang Raymond Choo**[4],
**Lianyong Qi**[5], **Xiaolong Xu**[6]*, **Amin Beheshti**[2], **Wanchun Dou**[1]

[1] State Key Laboratory for Novel Software Technology,
Department of Computer Science and Technology, Nanjing University, China
[2] School of Computing, Macquarie University, Australia
[3] School of Information and Physical Sciences, University of Newcastle, Australia
[4] Department of Information Systems and Cyber Security, University of Texas at San Antonio, USA
[5] College of Computer Science and Technology, China University of Petroleum (East China), China
[6] School of Computer and Software,
Nanjing University of Information Science and Technology, China

## Abstract

Federated learning (FL) has emerged as a prominent machine learning paradigm in edge computing environments, enabling edge devices to collaboratively optimize a global model without sharing their private data. However, existing FL frameworks suffer from efficacy deterioration due to the system heterogeneity inherent in edge computing, especially in the presence of domain shifts across local data. In this paper, we propose a heterogeneous FL framework *DapperFL*, to enhance model performance across multiple domains. In DapperFL, we introduce a dedicated Model Fusion Pruning (MFP) module to produce personalized compact local models for clients to address the system heterogeneity challenges. The MFP module prunes local models with fused knowledge obtained from both local and remaining domains, ensuring robustness to domain shifts. Additionally, we design a Domain Adaptive Regularization (DAR) module to further improve the overall performance of DapperFL. The DAR module employs regularization generated by the pruned model, aiming to learn robust representations across domains. Furthermore, we introduce a specific aggregation algorithm for aggregating heterogeneous local models with tailored architectures and weights. We implement DapperFL on a real-world FL platform with heterogeneous clients. Experimental results on benchmark datasets with multiple domains demonstrate that DapperFL outperforms several state-of-the-art FL frameworks by up to 2.28%, while significantly achieving model volume reductions ranging from 20% to 80%. Our code is available at: `https://github.com/jyzgh/DapperFL`.

## 1 Introduction

Federated Learning (FL), an emerging distributed machine learning paradigm in edge computing environments [1, 2], enables participant devices (i.e., clients) to optimize their local models while a central server aggregates these local models into a global model [3]. In contrast to traditional centralized machine learning paradigms, FL facilitates the collaborative training of a global model by distributed clients without the need for transmitting raw local data, thus mitigating privacy concerns associated with data transmission [2].

---

*Corresponding author: Xiaolong Xu (xlxu@nuist.edu.cn).

38th Conference on Neural Information Processing Systems (NeurIPS 2024).

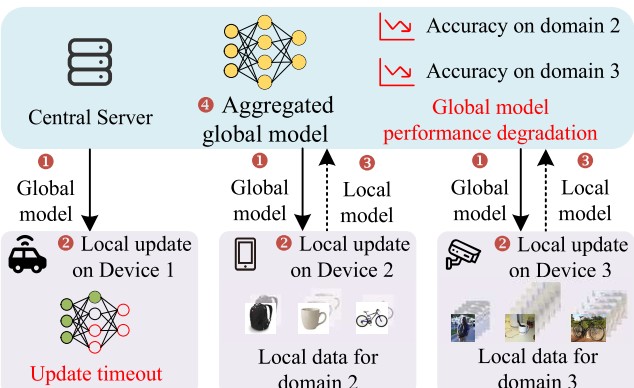

Figure 1: A motivational example of DapperFL with heterogeneous devices and multiple domains.

However, there are a number of challenges associated with FL deployments in edge computing settings, such as: **a) System heterogeneity.** Within prevalent FL environments, participant clients generally exhibit diverse and constrained system capabilities (e.g., CPU architecture, available memory, battery status, etc.). This heterogeneity often results in low-capability clients failing to complete local training of FL, consequently diminishing the performance of the aggregated global model [4, 5, 6]. **b) Domain shifts.** Owing to the distributed nature of FL, the data distributions among participant clients vary significantly [7, 8, 9]. Fig. 1 visually demonstrates the impact of these issues on FL performance, thus serving as the motivation for our work. Specifically, a) System heterogeneity: Device 1, with stringent resource constraints, fails to complete its model update within the maximum allowed time. As a result, the central server does not consider Device 1's model update, thus excluding it from the FL process and missing out on the data features captured by Device 1. b) Domain shifts: Devices 2 and 3 collect data from different domains, leading to diverse local models. With non-IID data, the global model aggregation may become biased, particularly if Device 3 contributes more data, thus potentially limiting the benefit for Device 2.

Several studies have been devoted to addressing these challenges while only addressing each challenge independently. In addressing the system heterogeneity challenge, existing FL frameworks [10, 11, 12] compress local models for heterogeneous clients, thereby enabling the inclusion of low-capability clients into the FL process. However, these frameworks typically assume that the local data on clients shares a uniform domain, and the compressed models are tailored for each individual client. Such compressed models are susceptible to over-fitting on their local domains due to the presence of domain shifts in practice. Consequently, the central server struggles to aggregate a robust global model with strong Domain Generalization (DG) capabilities. To address the domain shifts challenge, existing solutions exemplified by works such as [13, 14, 15], explore domain-invariant learning approaches either within the centralized machine learning paradigm or within ideal FL environments. However, these solutions unrealistically neglect the resource consumption of the clients, rendering them unsuitable for direct application in heterogeneous FL. Moreover, the inherent nature of FL yields non-independent and identically distributed (non-IID) data across the clients, further impeding these solutions from learning a domain-invariant global model, owing to limited data samples or labels available on individual clients. Despite existing work devoted to independently addressing system heterogeneity or domain shifts, few of them tackle these challenges simultaneously.

To bridge the gaps observed in existing studies, we propose DapperFL, an innovative FL framework designed to enhance model performance across multiple domains within heterogeneous FL environments. DapperFL addresses the system heterogeneity challenge through the deployment of a dedicated Model Fusion Pruning (MFP) module. The MFP module generates personalized compact local models for clients by pruning them with fused knowledge derived from both the local domain and remaining domains, thus ensuring domain generalization of the local models. Additionally, we introduce a Domain Adaptive Regularization (DAR) module to improve the overall performance of DapperFL. The DAR module segments the pruned model into an encoder and a classifier, intending to encourage the encoder to learn domain-invariant representations. Moreover, we propose a novel model aggregation approach tailored to aggregating heterogeneous local models with varying architectures and weights. We summarize our contributions as follows:

- We propose the MFP module to prune local models with personalized footprints leveraging both domain knowledge from local data and global knowledge from the server, thereby addressing the system heterogeneity issue in the presence of domain shifts. Additionally, we introduce a heterogeneous aggregation algorithm for aggregating the pruned models.

- We propose the DAR module to enhance the performance of the pruned models produced by the MFP module. The DAR module introduces a regularization term on the local objective to encourage clients to learn robust representations across various domains, thereby adaptively alleviating the domain shifts problem.

- We implement the proposed framework on a real-world FL platform and evaluate its performance on two benchmark datasets with multiple domains. The results demonstrate that DapperFL outperforms several state-of-the-art frameworks (i.e., [3, 16, 14, 15, 10, 17, 11, 12]) by up to 2.28%, while achieving adaptive model volume reductions on heterogeneous clients.

## 2  Related Work

**Heterogeneous Federated Learning.** In heterogeneous FL, diverse system capabilities and data distributions across clients often result in performance degradation of the global model [18, 19, 20]. Extensive studies have made efforts to address these heterogeneity issues through various solutions. For example, studies in [10, 21, 22, 11, 23, 12] adopt model sparsification techniques to reduce the volume of local models, thereby involving low-capability clients in the FL process. The studies in [17, 24, 25] leverage dedicated objective functions and specialized training steps to address the data heterogeneity issue, with studies in [17, 25] allowing clients to conduct varying numbers of local updates and consequently alleviating system heterogeneity issue. Several studies [5, 26, 27] selectively optimize or transmit a fraction of the local model's parameters to reduce computational or communication resource consumption. Studies in [28, 29, 30] split the model into several sub-models and offload a subset of sub-models to the server for updating, therefore alleviating the training burden of clients. However, these heterogeneous FL frameworks commonly assume the data heterogeneity (i.e., non-IID data) exclusively involves distribution shifts in the number of samples and/or labels, while neglecting the existence of domain shifts.

**Domain Generalization (DG).** DG is originally proposed in centralized machine learning paradigms to address the problem of domain shifts. Existing centralized studies assume access to the entire dataset during the model training process and propose various solutions to achieve DG [31]. For example, the studies in [32, 33, 34] focus on learning domain-invariant representations that can be generalized to unseen domains. In contrast to learning domain-invariant representations, several studies (e.g., [35, 36, 37, 38]) train a generalizable model across multiple domains leveraging meta-learning or transfer learning techniques. Additionally, there are studies (e.g., [39, 40, 13]) that focus on the characteristics of domains, enhancing the generality of the model by properly augmenting the style of domain or instance. Unfortunately, the fundamental assumption of centralized machine learning is not satisfied in FL, where the dataset is distributed among clients who are restricted from sharing their data. Despite a few recent studies exploring DG approaches in FL [41, 14, 15, 9] through representation learning, prototype learning, and/or contrastive learning techniques, they typically neglect the inherent system heterogeneity feature of FL. Consequently, these approaches have shown limited effectiveness in heterogeneous FL due to strict resource constraints. In contrast, we explored a distributed model pruning approach that leverages both the local domain knowledge and the global knowledge from all clients, thereby reducing resource consumption in heterogeneous FL with the presence of domain shifts. Additionally, we design a specific regularization technique for updating the pruned local models, thereby further enhancing model performance across domains.

## 3  DapperFL Design

### 3.1  Overview of DapperFL

DapperFL comprises two key modules, namely Model Fusion Pruning (MFP) and Domain Adaptive Regularization (DAR). The MFP module is designed to compress the local model while concurrently addressing domain shift problems, and the DAR module employs regularization generated by the compressed model to further mitigate domain shift problems and hence improve the overall perfor-

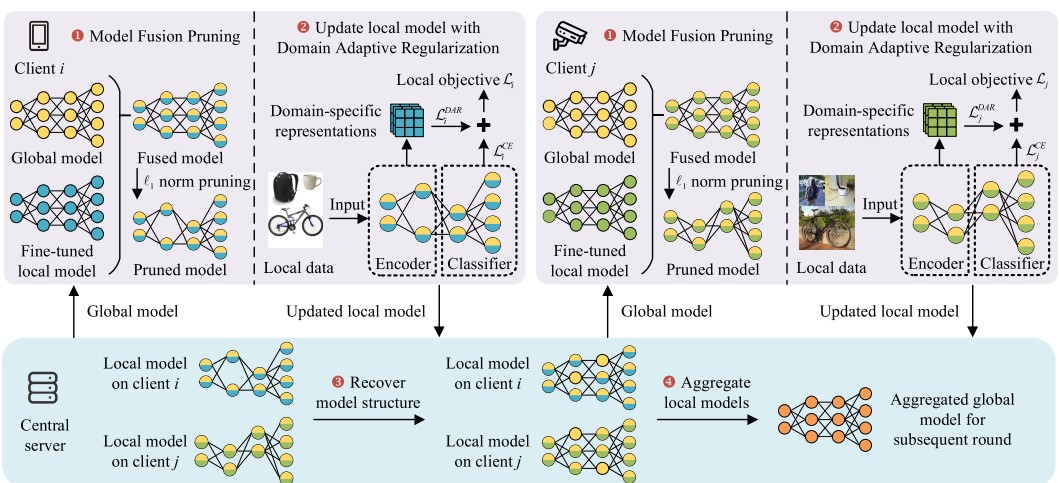

Figure 2: Overview of DapperFL framework with two clients for each communication round.

mance of DapperFL. Additionally, to handle the aggregation of heterogeneous models produced by the MFP module, we introduce a dedicated heterogeneous model aggregation algorithm.

Figure 2 explains the DapperFL's workflow during each communication round $t$, which is also described as follows. ① The MFP module within each client $i \in \mathcal{C}$ calculates the fusion model $\boldsymbol{w}_i^t$ using both the global and fine-tuned local models. The global model $\mathcal{W}^{t-1}$ is downloaded from the central server,[2] and the local model $\hat{\boldsymbol{w}}_i^t$ is fine-tuned on the client's local data with an initial epoch. Subsequently, the MFP calculates a binary mask matrix $\boldsymbol{M}_i^t$ using $\ell_1$ norm and produces the pruned local model $\boldsymbol{w}_i^t \odot \boldsymbol{M}_i^t$. ② The DAR module updates the pruned model $\boldsymbol{w}_i^t \odot \boldsymbol{M}_i^t$ over several epochs with a dedicated local objective $\mathcal{L}_i$. During local updating, the DAR module segments the local model into an encoder and a classifier, which are responsible for generating local representations and performing predictions, respectively. The local objective $\mathcal{L}_i$ is then constructed by combing regularization $\mathcal{L}_i^{DAR}$ of the local representations with the common cross-entropy loss $\mathcal{L}_i^{CE}$. Next, the client transmits the updated local model to the central server. ③ To aggregate local models with heterogeneous structures, the central server recovers the structure of received local models using the previous round's global model $\mathcal{W}^{t-1}$.[2] ④ The central server aggregates the recovered local models through weighted averaging, obtaining the global model $\mathcal{W}^t$ for round $t$.

## 3.2 Model Fusion Pruning on Edge Devices

In this subsection, we present the design of the MFP module employed by DapperFL. The goal of the MFP module is to tailor the footprint of the local model for edge devices in the presence of domain shifts in the local data, thereby addressing the system heterogeneity problem. Inspired by the spirit of the transfer learning [42, 43, 44], MFP fuses the global model $\mathcal{W}^{t-1}$ into the fine-tuned local model $\hat{\boldsymbol{w}}_i^t$ to learn the cross-domain knowledge. This avoids over-fitting the model to the local domain while enhancing the generality of the model. After that, MFP calculates a binary mask matrix $\boldsymbol{M}_i^t$ to generate the pruned local model. The detailed pruning process is described in Algorithm 1.

Specifically, in the initial epoch of each communication round $t \in [1, T]$, the MFP module first fine-tunes the global model $\mathcal{W}^{t-1}$ on local data $\mathcal{D}_i$ to produce local model $\hat{\boldsymbol{w}}_i^t$ (in line 1). We utilize one epoch of local training to determine the pruned models for the following reasons: 1) Additional local epochs do not significantly enhance the model's performance, which justifies the use of a single epoch for efficiency. As noted in [45], experiments have demonstrated that extending local training beyond one epoch yields results comparable to those achieved with just one epoch. 2) In previous domain generalization-related FL research, such as [45], one epoch is also employed to collect local domain information. This method has proven adequate for capturing essential features and domain characteristics. 3) Pioneering research in model design and neural architecture search, such as [46],

---

[2]In the first communication round, the initial backbone model is used as the global model.

---

**Algorithm 1** Model Fusion Pruning of DapperFL

---

**Input:** Global model $\mathcal{W}^{t-1}$, local data $\mathcal{D}_i$, pruning ratio $\rho_i$
**Output:** Pruned local model $\boldsymbol{w}_i^t \odot \boldsymbol{M}_i^t$
1: $\hat{\boldsymbol{w}}_i^t \leftarrow$ Fine-tune global model $\mathcal{W}^{t-1}$ on local data $\mathcal{D}_i$
2: $\boldsymbol{w}_i^t \leftarrow$ Fuse the global model $\mathcal{W}^{t-1}$ into the local model $\hat{\boldsymbol{w}}_i^t$ using Eq. 1 and Eq. 2
3: $\boldsymbol{M}_i^t \leftarrow$ Calculate binary mask matrix by $\ell_1$ norm with pruning ratio $\rho_i$
4: $\boldsymbol{w}_i^t \odot \boldsymbol{M}_i^t \leftarrow$ Prune the local model $\boldsymbol{w}_i^t$ with binary mask matrix $\boldsymbol{M}_i^t$
5: **return** Pruned local model $\boldsymbol{w}_i^t \odot \boldsymbol{M}_i^t$

---

has demonstrated that a few epochs are sufficient to obtain a coarse estimate of a sub-model. This approach is effective in quickly assessing model configurations without extensive computation.

Next, the MFP module fuses the global model $\mathcal{W}^{t-1}$ into the local model $\hat{\boldsymbol{w}}_i^t$ to embed the cross-domain information into the local model (in line 2). The fusion operation is formulated as follows:

$$\boldsymbol{w}_i^t = \alpha^t \mathcal{W}^{t-1} + (1 - \alpha^t)\hat{\boldsymbol{w}}_i^t, \tag{1}$$

where $\alpha^t$ is the fusion factor used to control the quality of the fused global model $\mathcal{W}^{t-1}$. Considering that the local data on the client is typically limited, resulting to updating at the beginning of FL requires more guidance from the global model by exploring the commonalities across different domains. As the FL process proceeds, the local model is capable of learning more domain-dependent information from itself. Thus, we design a dynamic adjusting mechanism to modify the $\alpha^t$ during the FL training. The $\alpha^t$ is initialized with a relatively large value $\alpha_0$ and decreases as the FL process proceeds. The decreased speed is controlled by a sensitivity factor $\epsilon$. We assign a minimum value $\alpha_{min}$ for $\alpha^t$ to ensure the local model will consistently learn the knowledge from other domains. Formally, the dynamic adjusting mechanism for $\alpha^t$ is described as follows:

$$\alpha^t = \max\{(1 - \epsilon)^{t-1}\alpha_0, \alpha_{min}\}. \tag{2}$$

Subsequently, the MFP module calculates a binary mask matrix $\boldsymbol{M}_i^t \in \{0, 1\}^{|\boldsymbol{w}_i^t|}$ for pruning the fused local model $\boldsymbol{w}_i^t$ (in line 3). The matrix $\boldsymbol{M}_i^t$ is derived through the channel-wise $\ell_1$ norm, which has proven to be effective and efficient in assessing the importance of parameters [47, 48]. The elements in $\boldsymbol{M}_i^t$ with a value of "0" indicate the corresponding parameters that need to be pruned, while those with a value of "1" indicate the parameters that will be retained. The pruning ratio $\rho_i$ determines the proportion of "0" in $\boldsymbol{M}_i^t$.[3] Finally, the MFP module prunes the local model $\boldsymbol{w}_i^t$ with the binary mask matrix $\boldsymbol{M}_i^t$ (in line 4). The pruned model can be represented as $\boldsymbol{w}_i^t \odot \boldsymbol{M}_i^t$. It is noteworthy that, the pruning strategy used in our DapperFL is structural pruning strategy (channel pruning), which is a hardware-friendly approach that can be easily implemented with popular machine learning libraries such as PyTorch, making it particularly suitable for deployment on edge devices.

### 3.3 Domain Adaptive Regularization

In FL, each client $i \in \mathcal{C}$ possess private local data $\mathcal{D}_i = \{x_i, y_i\}^{N_i}$, where $x \in \mathcal{X}$ denotes the input, $y \in \mathcal{Y}$ denotes the corresponding label, and $N_i$ represents the local data sample size. The data distribution $p_i(x, y)$ of client $i$ typically varies from that of other clients, i.e., $p_i(x) \neq p_j(x), p_i(x|y) \neq p_j(x|y)$, leading to the domain shifts problem. Due to the existence of domain shifts, representation $z_i$ generated by the local encoder varies among different clients, resulting in degraded prediction results of the local predictor. To address the domain shifts problem, we design a DAR module to enhance the performance of DapperFL across multiple domains while maintaining compatibility with the MFP module.

Specifically, the DAR module introduces a regularization term to the local objective to alleviate the bias of representations $z_i$ on different clients adaptively. To achieve this goal, we first segment each pruned local model $\boldsymbol{w} \odot \boldsymbol{M}$ into two parts, i.e., an encoder $\boldsymbol{w}_e \odot \boldsymbol{M}_e$ and a predictor $\boldsymbol{w}_p \odot \boldsymbol{M}_p$, where

---

[3]Following the conventional setting in heterogeneous FL [21, 11, 5, 12], we make the fundamental assumption that the system capabilities of the devices are available to the server and the pruning ratios for all devices are appropriately determined according to the system information.

$\boldsymbol{w} = \{\boldsymbol{w}_e, \boldsymbol{w}_p\}$ and $\boldsymbol{M} = \{\boldsymbol{M}_e, \boldsymbol{M}_p\}$.[4] The encoder is responsible for learning a representation $z_i$ given an input $x_i$, denoted as $z_i = g_e(\boldsymbol{w}_e \odot \boldsymbol{M}_e; x_i)$. While the predictor is responsible for predicting label $\hat{y}_i$ given the representation $z_i$, denoted as $\hat{y}_i = g_p(\boldsymbol{w}_p \odot \boldsymbol{M}_p; z_i)$. Subsequently, we construct a regularization term $\mathcal{L}^{DAR}$ on the local objective as follows:

$$\mathcal{L}_i^{DAR} = ||g_e(\boldsymbol{w}_e \odot \boldsymbol{M}_e; x_i)||_2^2. \tag{3}$$

Here, $|| \cdot ||_2^2$ represents the squared $\ell_2$ norm of the local representation. The $\ell_2$ norm implicitly encourages different local encoders to generate aligned robust representations adaptively, thereby mitigating the impact of domain shifts. We adopt the squared $\ell_2$ norm to construct the regularization term for the following reasons: a) The $\ell_2$ norm encourages each element of the representation to converge to 0 but not equal to 0. This property is advantageous compared to the $\ell_1$ norm, which tends to make smaller representation elements exactly equal to 0. Thus, the squared $\ell_2$ norm ensures that the pruned model retains more information in the representations. b) Higher order regularization introduces significant computational overhead compared to the $\ell_2$ norm, without significantly improving the regularization effectiveness.

Next, the cross-entropy loss used in the DAR module is constructed as:

$$\mathcal{L}_i^{CE} = -\frac{1}{|\mathcal{K}_i|} \sum_{k \in \mathcal{K}_i} y_{i,k} \log(\hat{y}_{i,k}), \tag{4}$$

where $\mathcal{K}_i$ denotes the set of possible labels on the client $i$, $\hat{y}_{i,k}$ denotes predicting label, and $y_{i,k}$ denotes ground-truth label. Finally, the training objective of each client is calculated as follows:

$$\mathcal{L}_i = \mathcal{L}_i^{CE} + \gamma \mathcal{L}_i^{DAR}, \tag{5}$$

where the $\gamma$ is a pre-defined coefficient controlling the importance of $\mathcal{L}_i^{DAR}$ relative to $\mathcal{L}_i^{CE}$.

## 3.4 Heterogeneous Model Aggregation

Despite the MFP module and the DAR module being capable of alleviating domain shift issues, the heterogeneous local models generated by the MFP module cannot be aggregated directly using popular aggregation algorithms. Therefore, we propose a specific FL aggregation algorithm for DapperFL to effectively aggregate these heterogeneous local models.

To preserve specific domain knowledge while transferring global knowledge to the local model, the central server first recovers the structure of local models before aggregating. Specifically, in each communication round $t$, the pruned local model is recovered as follows:

$$\boldsymbol{w}_i^t := \underbrace{\boldsymbol{w}_i^t \odot \boldsymbol{M}_i^t}_{\text{local knowledge}} + \underbrace{\mathcal{W}^{t-1} \odot \overline{\boldsymbol{M}}_i^t}_{\text{global knowledge}}, \tag{6}$$

where $\mathcal{W}^{t-1}$ is the global model aggregated at the $(t-1)$-th round, and $\overline{\boldsymbol{M}}_i^t$ denotes the logical NOT operation applied to $\boldsymbol{M}_i^t$. The first term $\boldsymbol{w}_i^t \odot \boldsymbol{M}_i^t$ contains local knowledge[5] specific to client $i$, while the second term $\boldsymbol{w}^{t-1} \odot \overline{\boldsymbol{M}}_i^t$ contains the global knowledge[6] from all clients. Additionally, the structure of $\boldsymbol{w}_i^t \odot \boldsymbol{M}_i^t$, which includes local knowledge, complements $\boldsymbol{w}_i^t \odot \boldsymbol{M}_i^t$. Consequently, the structure recovery operation not only reinstates the pruned model's architecture but also transfers global knowledge to the pruned model, which is essential for the subsequent aggregation process. By combining these two forms of knowledge, we aim to leverage both the specialized insights of local models and the generalized capabilities of the global model, thereby enhancing the performance and adaptability of DapperFL.

---

[4]We omit the client index $i$ and the communication round index $t$ for notation simplicity. In this work, all layers except the final linear layer act as the encoder, while the last linear layer of the model acts as the predictor.

[5]In this work, "local knowledge" refers to the feature extraction capabilities of the local model, which are learned from the specific data available in its local domain. This knowledge encapsulates the nuances and characteristics of the data that the local model has been trained on.

[6]The "global knowledge" represents the aggregated feature extraction capabilities of the global model, which are informed by data across all participating domains. The global model synthesizes diverse knowledge from different local models to provide a more generalized understanding that is applicable across multiple domains.

---

**Algorithm 2** DapperFL

---

**Input:** Initial global model $\mathcal{W}^0$, total number of communication rounds $T$, total number of local epochs $E$, local dataset $\mathcal{D}_i$ and pruning ratio $\rho_i$ for each client $i$
**Output:** Optimized global model $\mathcal{W}$
    **Local Updates:**
 1: $\boldsymbol{w}_i^t \odot \boldsymbol{M}_i^t \leftarrow$ Prune local model by the pruning ratio $\rho_i$ using Algorithm 1   // local epoch $e = 1$
 2: **for** local epoch $e \in [2, E]$ **do**
 3:    Update the pruned local model $\boldsymbol{w}_i^t \odot \boldsymbol{M}_i^t$ using local dataset $\mathcal{D}_i$ and local objective Eq.5
 4: **end for**
    **Server Executes:**
 5: **for** communication round $t \in [1, T]$ **do**
 6:    Send global model $\mathcal{W}^{t-1}$ to all clients   // waiting for the clients to finish updating
 7:    Receive updated models $\boldsymbol{w}_i^t \odot \boldsymbol{M}_i^t$ from all clients
 8:    $\boldsymbol{w}_i^t \leftarrow$ Recover the local models using Eq.6
 9:    $\mathcal{W}^t \leftarrow$ Aggregate the local models using Eq.7
10: **end for**
11: **return** Optimized global model $\mathcal{W}^t$

---

Finally, the global model is calculated by aggregating the recovered local models as follows:

$$\mathcal{W}^t = \sum_{i \in \mathcal{C}} \frac{|\mathcal{D}_i|}{|\mathcal{D}|} \boldsymbol{w}_i^t, \tag{7}$$

where $|\mathcal{D}_i|$ is the sample number in the local dataset on client $i$, and $|\mathcal{D}|$ is the total number of samples in the entire FL system. Benefiting from the heterogeneous aggregation algorithm, the aggregated global model retains knowledge from all clients, enabling it to perform robustly across domains.

The overall learning process of our proposed DapperFL is described in Algorithm 2.

## 4 Experimental Evaluation

### 4.1 Experimental Setup

**Implementation Details.** We implement DapperFL with a real-world FL platform FedML [49] with deep learning tool PyTorch [50]. We build the FL environment with a client set $\mathcal{C}$ containing 10 heterogeneous edge devices and a central server on the FedML platform. Following a convention setting [5, 25, 51], we categorize these heterogeneous devices into 5 levels according to their system capabilities, i.e., $\mathcal{C}_l \in \mathcal{C}$ ($l \in [1, 5]$), where $\mathcal{C}_l$ represents device set belongs to level $l$. The system capabilities of devices in set $\mathcal{C}_l$ decrease as level $l$ increases. Our experiments are conducted on a GPU server with 2 NVIDIA RTX 3080Ti GPUs. Each experiment is executed three times with three fixed random seeds to calculate average metrics and ensure the reproducibility of our results.

**Datasets and Data Partition.** We evaluate DapperFL on two domain generalization benchmarks, Digits [52, 53, 54, 55] and Office Caltech [56] that are commonly used in the literature for domain generalization. The Digits consists of the following four domains: MNIST, USPS, SVHN, and SYN. The Office Caltech consists of the following four domains: Caltech, Amazon, Webcam, and DSLR. For each benchmark, we distribute the simulated 10 clients randomly to each domain while guaranteeing that each domain contains at least one client and that each client only belongs to one domain. To generate non-IID local data, we randomly extract a proportion of data from the corresponding domain as the statistical characteristics vary among domains. Following the conventional data partition setting [15], we set 1% as the local data proportion of Digits and 20% as that in Office Caltech based on the complexity and scale of the benchmarks.

**Models.** We adopt ResNet10 and ResNet18 [57] as the backbone models for Digits and Office Caltech, respectively. In the following evaluations, both the DapperFL and comparison frameworks train the models from scratch for a fair comparison.

**Comparison Frameworks.** We compare DapperFL with 8 state-of-the-art (SOTA) FL frameworks, including FedAvg [3], MOON [16], FedSR [14], FPL [15], FedDrop [10], FedProx [17], FedMP [11],

Table 1: Comparison of model accuracy (%) on Digits.

| FL frameworks | System Heter. | MNIST | USPS | SVHN | SYN | Global accuracy |
|---|---|---|---|---|---|---|
| FedAvg [3] | ✗ | 95.89(1.47) | 86.84(0.80) | 78.39(3.24) | 33.63(2.87) | 71.81(0.46) |
| MOON [16] | ✗ | 93.03(1.97) | 78.38(5.81) | 84.45(7.55) | 25.97(3.28) | 69.44(0.53) |
| FedSR [14] | ✗ | 96.77(0.73) | 86.15(2.38) | 81.48(1.77) | 31.64(0.40) | 73.89(0.57) |
| FPL [15] | ✗ | 95.54(1.78) | 87.69(0.98) | 83.74(4.26) | 34.73(1.53) | 74.17(0.95) |
| FedDrop [10] | ✓ | 89.48(2.56) | 82.51(1.17) | 72.98(0.83) | 29.35(1.97) | 66.85(0.93) |
| FedProx [17] | ✓ | 96.68(0.96) | 83.96(0.73) | 76.69(3.50) | 30.95(1.42) | 70.74(0.52) |
| FedMP [11] | ✓ | 94.16(3.32) | 85.30(2.66) | 81.37(1.92) | 35.12(2.00) | 72.29(0.89) |
| NeFL [12] | ✓ | 84.98(1.07) | 88.49(4.17) | 78.41(2.33) | 36.02(5.72) | 67.64(0.30) |
| **DapperFL (ours)** | ✓ | 96.25(2.10) | 86.30(1.24) | 82.45(1.72) | 37.26(2.71) | **74.30(0.26)** |

Table 2: Comparison of model accuracy (%) on Office Caltech.

| FL frameworks | System Heter. | Caltech | Amazon | Webcam | DSLR | Global accuracy |
|---|---|---|---|---|---|---|
| FedAvg [3] | ✗ | 66.07(2.46) | 76.84(3.18) | 65.52(4.98) | 56.67(1.98) | 64.54(1.10) |
| MOON [16] | ✗ | 65.62(3.74) | 75.79(1.69) | 72.41(2.63) | 53.33(1.93) | 61.86(0.79) |
| FedSR [14] | ✗ | 62.95(2.25) | 78.95(3.29) | 75.86(3.59) | 50.00(3.34) | 65.47(1.13) |
| FPL [15] | ✗ | 63.84(3.17) | 82.63(4.11) | 65.52(2.63) | 60.00(3.85) | 65.45(1.15) |
| FedDrop [10] | ✓ | 66.07(0.89) | 79.47(2.30) | 56.90(3.98) | 53.33(6.94) | 60.58(1.42) |
| FedProx [17] | ✓ | 61.61(4.09) | 71.05(4.98) | 68.97(4.98) | 46.67(1.93) | 62.08(1.11) |
| FedMP [11] | ✓ | 65.62(2.49) | 75.79(2.43) | 56.90(3.59) | 66.67(3.34) | 62.34(0.93) |
| NeFL [12] | ✓ | 54.91(1.57) | 71.05(1.61) | 77.59(4.56) | 66.67(3.85) | 62.26(1.34) |
| **DapperFL (ours)** | ✓ | 64.73(1.03) | 81.58(3.29) | 74.14(1.99) | 66.67(3.85) | **67.75(0.97)** |

and NeFL [12]. These FL frameworks are either classical or focus on addressing system heterogeneity or domain shifts issues. The comparison frameworks are described in detail in Appendix A.

**Default Hyper-parameters.** To conduct fair evaluations, the global settings and local training hyper-parameters of FL are set to identical values to both the DapperFL and comparison FL frameworks. For the framework-specific hyper-parameters, we use their default settings without changing them. The hyper-parameters used in our evaluations are described in Appendix B.

**Evaluation Metrics.** In this paper, we evaluate the performance of the global model using the average Top-1 accuracy across all domains. In evaluating the resource consumption of the clients, we adopt both the total number of parameters and the Floating-Point Operations (FLOPs) of the local model.

## 4.2 Performance Comparison

**Model Performance Across Domains.** Tables 1 and 2 respectively provide detailed analyses of model accuracy on the Digits and Office Caltech benchmarks. The term "System Heter." indicates whether the respective framework supports system heterogeneity.[7] The results are reported as model accuracy with corresponding standard deviations in brackets.

As shown in both Tables 1 and 2, DapperFL achieves the highest global accuracy on both the Digits and Office Caltech benchmarks compared with the comparison frameworks. On Digits and Office Caltech, DapperFL's global accuracy is 0.13% and 2.28% better than the runner-up, respectively. This indicates that the global model learned by DapperFL is more robust across all domains and therefore possesses better domain generalization. Moreover, DapperFL always achieves competing accuracy on individual domains and even the best domain accuracy on the SYN, Amazon, and DSLR. This demonstrates that the DAR module of DapperFL implicitly encourages the encoder to learn domain-invariant representations, resulting in DapperFL being unlikely to over-fit on the specific

---

[7]Note that when evaluating frameworks that do not support heterogeneous clients, we allow all clients to participate in FL by disregarding resource constraints on clients.

domain(s) and possessing better domain generalization ability. Furthermore, DapperFL tolerates heterogeneous clients through personalized pruning ratio $\rho$, achieving adaptive resource consumption reductions of $\{20\%, 40\%, 60\%, 80\%\}$ for clients in $\{\mathcal{C}_2, \mathcal{C}_3, \mathcal{C}_4, \mathcal{C}_5\}$, while outperforming comparison FL frameworks that support heterogeneous clients. This is evidence that the MFP module of DapperFL effectively reduces the resource consumption of clients with domain shifts. In addition, we provide learning curves for both global and domain accuracies in Appendix C for a comprehensive analysis.

**Impact of Model Footprints.** In heterogeneous FL, resource constraints on the clients often result in limited model footprints (i.e., number of parameters and FLOPs). To investigate the impact of the model footprints on model performance, we compare the DapperFL with two SOTA FL frameworks (i.e., FedMP and NeFL) that are equipped with adaptive pruning techniques. The pruning ratio $\rho$ determines both the reduction in the number of parameters and the FLOPs. We maintain a consistent $\rho$ for every device, with predetermined values selected from the set $\{0.2, 0.4, 0.6, 0.8\}$.

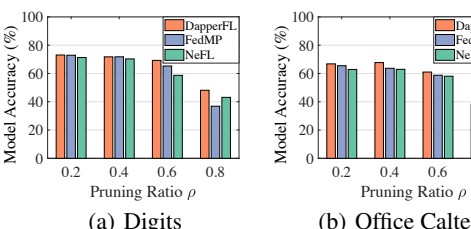

(a) Digits      (b) Office Caltech

Figure 3: Comparison of model accuracy of FedMP, NeFL, and DapperFL with different pruning ratios on the Digits and Office Caltech.

Figure 3 illustrates the comparative results under varying pruning ratios on both Digits and Office Caltech. As depicted, these frameworks tend to achieve higher accuracy with smaller pruning ratios, and DapperFL consistently outperforms the others across all $\rho$ values on both benchmarks. This underscores the superior adaptability of DapperFL in heterogeneous FL environments. Notably, in Figure 3(b), the accuracy of DapperFL at $\rho = 0.4$ surpasses its accuracy at $\rho = 0.2$. This discrepancy arises from the over-parameterization of ResNet18 for the classification task in Office Caltech. The improved accuracy observed at higher pruning rates suggests that the MFP module of DapperFL effectively prunes unnecessary parameters while enhancing model generalization rather than compromising it. This also demonstrates the reason behind DapperFL's superiority over comparison frameworks that employ full-size models. Furthermore, we comprehensively evaluate the effect of DapperFL's pruning ratio $\rho$ on model performance in Appendix D.

## 4.3 Ablation Study

**Effect of Key Modules in DapperFL.** We evaluate the performance of DapperFL with and without the MFP and DAR modules, individually and in combination. The following configurations are considered: "DapperFL w/o MFP+DAR", DapperFL without the MFP and DAR modules. "DapperFL w/o DAR", DapperFL without the DAR module. "DapperFL w/o MFP", DapperFL without the MFP module. "DapperFL", the complete DapperFL framework. It is noteworthy that when the MFP module is not employed in DapperFL, it performs model pruning using $\ell_1$ norm on the local models directly.

Table 3: Effect of DapperFL's Key Modules on Digits and Office Caltech.

| Configuration | Digits | Office |
|---|---|---|
| DapperFL w/o MFP+DAR | 71.94% | 62.65% |
| DapperFL w/o DAR | 72.37% | 64.88% |
| DapperFL w/o MFP | 73.34% | 66.28% |
| DapperFL | **74.30%** | **67.75%** |

Table 3 summarizes the ablation study results on both benchmarks. On Digits, the results indicate a gradual improvement in accuracy as we incorporate the MFP and DAR modules into DapperFL. DapperFL achieves the highest accuracy at 74.30%, highlighting the synergistic effect of both MFP and DAR in enhancing model performance. Similarly, we observe a consistent pattern on Office Caltech. DapperFL w/o MFP+DAR yields the lowest accuracy, while the inclusion of both MFP and DAR results in a steady improvement. The complete DapperFL framework attains the highest accuracy at 67.75%, reinforcing the complementary roles played by MFP and DAR in bolstering domain adaptability in heterogeneous FL.

**Effect of Hyper-Parameters within the MFP Module.** We investigate the influence of the hyper-parameters $\alpha_0$, $\alpha_{min}$, and $\epsilon$ within the MFP module on the overall performance of DapperFL. In this experiment, only the hyper-parameter under evaluation is modified, while others retain their default settings as outlined in Appendix B.

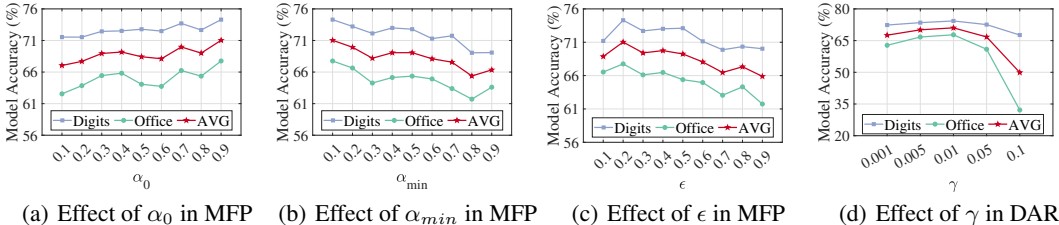

| (a) Effect of $\alpha_0$ in MFP | (b) Effect of $\alpha_{min}$ in MFP | (c) Effect of $\epsilon$ in MFP | (d) Effect of $\gamma$ in DAR |

Figure 4: The effect of hyper-parameters in the MFP and DAR modules on model accuracy. "Office" in the legend represents the Office Caltech benchmark, and the average accuracy calculated for both benchmarks is labeled as "AVG".

Figure 4(a) illustrates the effect of $\alpha_0$ on model accuracy. The $\alpha_0$ parameter dictates the initial weight of the global model fused into the local model. In both benchmarks, the optimal $\alpha_0$ is evident at 0.9, resulting in the highest average accuracy. This result underscores the significance of infusing the local model with knowledge from other domains early in the FL process to enhance its generalization. Figure 4(b) describes the effect of $\alpha_{min}$ on model accuracy, where $\alpha_{min}$ determines the minimum weight of the global model fused into the local model. In both benchmarks, the optimal average accuracy is achieved when $\alpha_{min}$ is set to 0.1. This finding signifies that, as the FL process proceeds, the local model benefits from acquiring more domain-specific knowledge to uphold its personalization. Figure 4(c) explores the effect of $\epsilon$ on model accuracy. The $\epsilon$ parameter controls the rate at which the fusion factor $\alpha_0$ diminishes towards $\alpha_{min}$. In both benchmarks, the optimal $\epsilon$ is discernible at 0.2, resulting in the highest average accuracy. This observation emphasizes that the most effective rate of decrease for the fusion factor $\alpha_0$ towards $\alpha_{min}$ is achieved when $\epsilon = 0.2$ in both benchmarks. Furthermore, recognizing the abnormal trends in the relationship between model accuracy and hyper-parameter $\epsilon$ when it is less than 0.2, we implement Bayesian search as an automatic selection mechanism to find the optimal value of $\epsilon$. The results are provided in Appendix E.

**Effect of Hyper-Parameter within the DAR Module.** Figure 4(d) provides configurations with varying $\gamma$ values across both Digits and Office Caltech. The results show an increase in accuracy with higher $\gamma$ values until $\gamma = 0.01$ on Digits, where the highest accuracy of 74.30% is achieved. A notable drop in accuracy is observed at $\gamma = 0.1$, indicating that excessively large regularization may hinder model performance on Digits. Similar to Digits, an increase in accuracy is observed with higher $\gamma$ values until $\gamma = 0.01$, reaching the highest accuracy of 67.75%. The impact of regularization is more pronounced on Office Caltech, as seen by the significant drop in accuracy at $\gamma = 0.1$. This suggests that the regularisation factor $\gamma$ in the DAR Module plays a crucial role in solving the domain shift problem in FL and a careful choice of $\gamma$ can improve the model performance.

## 5    Conclusion

We have described our proposed FL framework DapperFL tailored for heterogeneous edge devices with the presence of domain shifts. Specifically, DapperFL integrates a dedicated MFP module and a DAR module to achieve domain generalization adaptively in heterogeneous FL. The MFP module addresses system heterogeneity challenges by shrinking the footprint of local models through personalized pruning decisions determined by both local and remaining domain knowledge. This effectively mitigates limitations associated with system heterogeneity. Meanwhile, the DAR module introduces a specialized regularization technique to implicitly encourage the pruned local models to learn robust local representations, thereby enhancing DapperFL's overall performance. Furthermore, we designed a specific aggregation algorithm for DapperFL to aggregate heterogeneous models produced by the MFP module, aiming to realize a robust global model across multiple domains. The evaluation of DapperFL's implementation on a real-world FL platform demonstrated that it outperforms several SOTA FL frameworks on two benchmark datasets comprising various domains.

**Limitations and Future Work.** Despite its potential in addressing system heterogeneity and domain shifts, DapperFL introduces four hyper-parameters $\alpha_0$, $\alpha_{min}$, $\epsilon$, and $\gamma$ associated with the DG performance of the global model. A potential future direction involves the automatic selection of these hyper-parameters, therefore enhancing the flexibility and accessibility of the DapperFL.

## Acknowledgments and Disclosure of Funding

This research is supported part by the National Natural Science Foundation of China No. 92267104 and No. 62372242. Dr. Xuyun Zhang is the recipient of an ARC DECRA (project No. DE210101458) funded by the Australian Government. The work of K.-K. R. Choo is supported only by the Cloud Technology Endowed Professorship.

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

# A Introduction to Comparison Frameworks

The comparison FL frameworks used in this work are introduced as follows:

**FedAvg** [3]: FedAvg is a classical FL framework that is widely adopted to aggregate distributed local models. During the aggregation period of FedAvg, the clients upload the updates to the server, and the central server is responsible for generating the global model by performing weighted averaging on these local updates.

**FedDrop** [10]: FedDrop reduces the computational overhead of local updating and communication costs by generating compact models for clients. In FedDrop, the server adopts lossy compression techniques to compress the global model with a uniform compression ratio for all clients. To satisfy the resource requirements of low-level clients, FedDrop needs to choose a large pruning ratio to compress the global model.

**FedProx** [17]: FedProx introduces a proximal term into the objective function, aimed at incentivizing participants to uphold similarity with the global model throughout local training, meanwhile affording low-capacity clients the latitude to execute fewer local updates.

**MOON** [16]: MOON is a simple yet effective FL framework tailored for non-IID local data among clients. It capitalizes on model representation similarities to enhance the local training of individual clients through model-level contrastive learning.

**FedMP** [11]: FedMP employs an adaptive local model pruning mechanism in each FL communication round to address the diverse resource constraints present on client devices. Simultaneously, it leverages a Residual Recovery Synchronous Parallel (R2SP) scheme to aggregate models from heterogeneous clients.

**FedSR** [14]: FedSR incorporates two regularization techniques to implicitly align representations across domains in the FL environments. These regularizers implicitly align the marginal and conditional distributions of the representations, thereby enhancing domain generalization.

**NeFL** [12]: NeFL employs a combination of depthwise and widthwise scaling techniques to partition a model into submodels. Its primary objective is to enable resource-constrained clients to seamlessly engage in the FL process, meanwhile facilitating the training of models on more extensive datasets. To mitigate the challenges associated with training multiple submodels featuring varying architectures, NeFL introduces a parameter decoupling mechanism.

Table 4: Default hyper-parameters used in our evaluations.

| Type | Hyper-parameter | Default value |
|---|---|---|
| Global settings of FL | Communication round | 100 |
| | Client number | 10 |
| | Client participation rate | 100% |
| Local training of FL | Local epoch | 5 |
| | Batch size | 64 |
| | Learning rate | 0.01 |
| | Momentum | 0.9 |
| | Weight decay | $1 \times 10^{-5}$ |
| Framework-specific | $\mu$ in FedProx | 0.1 |
| | $\mu$ in MOON | 1 |
| | $\tau$ in MOON | 0.5 |
| | $\alpha^{L2R}$ in FedSR | 0.01 |
| | $\alpha^{CMI}$ in FedSR | 0.001 |
| | $\tau$ in FPL | 0.02 |
| | $\alpha_0$ in DapperFL | 0.9 |
| | $\alpha_{min}$ in DapperFL | 0.1 |
| | $\epsilon$ in DapperFL | 0.2 |
| | $\gamma$ in DapperFL | 0.01 |

**FPL** [15]: FPL aims to learn a robust global model in FL with domain shifts. It improves generalization while maintaining discriminability by leveraging contrastive learning with cluster prototypes. Additionally, FPL utilizes unbiased prototypes to provide a fair and stable convergence point.

## B   Default Hyper-parameters

The default hyper-parameters used in our evaluations are described in Table 4.

## C   Learning Curves

We provide learning curves of global accuracy and domain accuracy in Figure 5 and Figure 6, respectively. These curves provide the maximum Top-1 accuracy achieved until each communication round of FL.

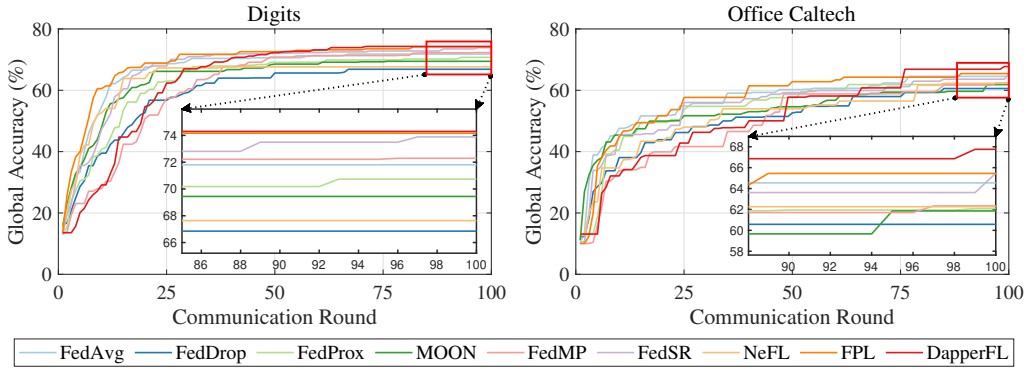

Figure 5: Learning curves of global accuracies for FedAvg, FedDrop, FedProx, MOON, FedMP, FedSR, NeFL, FPL, and DapperFL on the Digits benchmark and the Office Caltech benchmark.

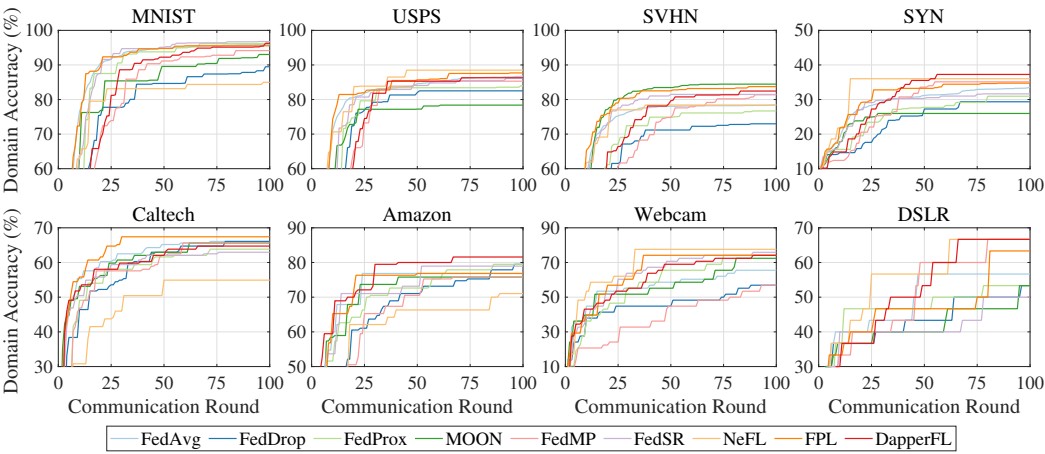

Figure 6: Learning curves of domain accuracies for FedAvg, FedDrop, FedProx, MOON, FedMP, FedSR, NeFL, FPL, and DapperFL on the Digits benchmark and the Office Caltech benchmark.

## D   Effect of Pruning Ratio $\rho$

We systematically investigate the impact of varying pruning ratios ($\rho$) on the model performance of DapperFL using both the Digits and Office Caltech benchmarks. Pruning ratios ranging from 0.2 to 0.8 are considered, representing different degrees of model compression. Table 5 presents the

model footprint and accuracy of DapperFL under different pruning ratios on the Digits benchmark. As the pruning ratio increases, the number of parameters (#Para) and FLOPs decrease, reflecting the expected reduction in model size and computational complexity. Notably, the model maintains competitive accuracy when pruning ratios increase from 0.2 to 0.6, indicating the robustness of DapperFL to varying degrees of model compression. Similarly, Table 6 details the model footprint and accuracy of DapperFL under different pruning ratios on the Office Caltech benchmark. Consistent with the Digits benchmark results, the model adapts effectively to increased pruning from 0.2 to 0.6, with reductions in both parameters and FLOPs.

In summary, DapperFL consistently outperforms the baseline frameworks that utilize the entire model, such as FedAvg and MOON, as illustrated in Table 1 and Table 2, when employing a pruned model with a pruning ratio $\rho$ ranging from 0.2 to 0.6. Despite incurring an inevitable accuracy loss at $\rho = 0.8$ on both benchmarks, DapperFL still outperforms FedMP and NeFL, both equipped with adaptive pruning capabilities, as illustrated in Figure 3. These observations collectively demonstrate the robustness of DapperFL across a spectrum of pruning ratios, emphasizing its potential for deployment in resource-constrained FL environments. The ability to achieve substantial model compression (with the pruning ratio $\rho$ ranging from 0.2 to 0.6) without significant loss in accuracy makes DapperFL a promising solution for real-world applications where resource efficiency is paramount.

Table 5: Model footprint and accuracy of DapperFL under varying pruning ratios on Digits.

| Pruning ratio $\rho$ | #Para | FLOPs | MNIST | USPS | SVHN | SYN | Global accuracy |
|---|---|---|---|---|---|---|---|
| 0.2 | 3.92M | 203.34M | 94.86% | 83.36% | 85.55% | 32.84% | 73.06% |
| 0.4 | 2.94M | 152.50M | 89.42% | 80.77% | 84.13% | 35.57% | 71.76% |
| 0.6 | 1.96M | 101.67M | 91.79% | 82.16% | 77.59% | 29.65% | 69.27% |
| 0.8 | 0.98M | 50.83M | 63.38% | 66.17% | 58.38% | 21.64% | 48.14% |

Table 6: Model footprint and accuracy of DapperFL under varying pruning ratios on Office Caltech.

| Pruning ratio $\rho$ | #Para | FLOPs | Caltech | Amazon | Webcam | DSLR | Global accuracy |
|---|---|---|---|---|---|---|---|
| 0.2 | 8.94M | 366.13M | 68.30% | 80.53% | 63.79% | 60.00% | 66.80% |
| 0.4 | 6.70M | 274.60M | 70.09% | 79.47% | 67.24% | 56.67% | 67.71% |
| 0.6 | 4.47M | 183.06M | 58.48% | 80.00% | 67.24% | 50.00% | 61.02% |
| 0.8 | 2.23M | 91.53M | 43.75% | 53.16% | 31.03% | 33.33% | 38.28% |

# E   Automatic Selection Mechanism for Hyper-parameter $\epsilon$

We run DapperFL on the Office Caltech benchmark 40 times, adopting a distinct $\epsilon$ of less than 0.2 each time. The values are selected using the Bayesian search. The results are presented in Fig. 7.

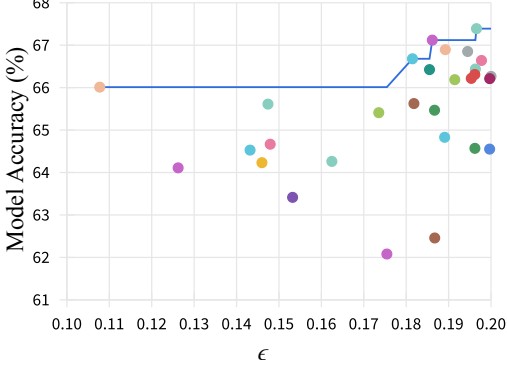

Figure 7: Model accuracy of DapperFL on the Office Caltech for different $\epsilon$ searched by the Bayesian algorithm. The maximum model accuracy as $\epsilon$ increases is plotted with the blue line for analysis.

As illustrated, the Bayesian search-based automatic selection mechanism indicates that model accuracy is likely to reach a higher level when $\epsilon$ approaches 0.2, aligning with our default setting of $\epsilon = 0.2$. It is noteworthy that, owing to the nature of Bayesian search, the sampled $\epsilon$ values are not uniformly distributed, as they tend to bias towards the optimal $\epsilon$ that maximizes model accuracy.

