# OpenReview forum: "DapperFL: Domain Adaptive Federated Learning with Model Fusion Pruning for Edge Devices"
_NeurIPS.cc/2024/Conference — NeurIPS 2024 oral_

### Official Review · Reviewer_npsw · 2024-07-10

**Soundness:** 3
**Presentation:** 3
**Contribution:** 4
**Rating:** 8
**Confidence:** 5

**Summary:**

This paper proposes DapperFL, a novel framework for domain adaptive federated learning designed specifically for heterogeneous edge devices. It addresses the challenge of system heterogeneity and domain shifts in FL by integrating a Model Fusion Pruning (MFP) module and a Domain Adaptive Regularization (DAR) module. The framework also incorporates a novel aggregation approach to handle heterogeneous local models with varying architectures and weights.

**Strengths:**

•	Originality: The author focuses on simultaneously addressing system heterogeneity and domain shifts in the context of federated learning. While previous work has devoted to address these two aspects individually, there is a lack of solutions that tackle both problems simultaneously. DapperFL offers a unique combination of model pruning and domain adaptation techniques within a federated learning context, addressing system heterogeneity and domain shifts in a novel way.

•	quality: The framework is rigorously developed, with a clear methodology and implementation details provided.

•	clarity:This paper's figures are clear and easy to understand, and the explanations of the framework(MFP, DAR, Aggregation) are well-organized and coherent.

**Weaknesses:**

•	The introduction of the two issues, system heterogeneity and domain shift, in this article are not very clear. Regarding system heterogeneity, the author places emphasis on low-capability clients being unable to perform local training, but this emphasis lacks relevance to the rest of the paper. Additionally, the example of the cameras in the domain shifts problem is somewhat biased towards the "multi-view task." For the issues of model heterogeneity and domain shift, the author needs to provide a clearer exposition of the application scenarios where these two problems intersect. If feasible, it would be beneficial to supplement the explanation with illustrations.

•	The strategy employed in the Model Fusion Pruning phase is not particularly innovative. Moreover, this strategy requires the storage of the entire global model and additional extensive computations under resource-constrained settings, which severely contradicts the problem that this paper aims to address. Furthermore, the pruning strategy employed in this article is the unstructured pruning strategy, so the actual contribution to the problem of system heterogeneity is limited.

•	Althought the framework diagram is already very clear, it is suggested to include model symbols corresponding to the algorithm in order to provide a clearer explanation of the model's changes at each step.

•	The method details in this article bear some resemblance to existing work, and there is a lack of references to these works. ( Diao, Enmao, Jie Ding, and Vahid Tarokh. "Heterofl: Computation and communication efficient federated learning for heterogeneous clients." arXiv preprint arXiv:2010.01264 (2020), Yuang Jiang, Shiqiang Wang, Bong Jun Ko, Wei-Han Lee, and Leandros Tassiulas. 2019. Model Pruning Enables Efficient Federated Learning on Edge Devices. CoRR abs/1909.12326 (2019). )

•	In addition, there are some approaches that address model heterogeneity and domain shift problems through distillation. Although this paper focuses on pruning paths, these methods should also be mentioned and even compared. (Zhu, Zhuangdi, Junyuan Hong, and Jiayu Zhou. "Data-free knowledge distillation for heterogeneous federated learning." International conference on machine learning. PMLR, 2021. Huang, Wenke, Mang Ye, and Bo Du. "Learn from others and be yourself in heterogeneous federated learning." Proceedings of the IEEE/CVF Conference on Computer Vision and Pattern Recognition. 2022.)

**Questions:**

•	Is there a practical difference between the fine-tuning global model steps inspired by the spirit of transfer learning in this article and the local model training after initializing the global model in regular federated learning? Can you provide a more specific explanation of the role of transfer learning in this step?

•	The global model deployment step appears to result in the global model being stored on the local client and requiring fine-tuning as well as mask operations for traversing each neural unit. Does this contradict resource-constrained scenarios in heterogeneous systems?

**Limitations:**

•	The authors pointed out the limitations of the paper in terms of hyper-parameters selection. However, this article still has certain limitations in its solution for system heterogeneity. In addition, it is necessary to add theoretical conversion analysis of the proposed federated learning algorithm.

---

> ### Author Rebuttal · Authors · 2024-08-06
>
> We sincerely thank you for your constructive and helpful comments. Below we address your concerns in order.
>
>
> ## Response to Weakness 1:
>
> **Clarification on System Heterogeneity:**
> In our paper, we emphasize that low-capability clients "fail to complete" local training rather than being "unable to perform" local training. Following the conventional setting [1,2,3], low-capability clients (i.e., stragglers) possess the basic capability to train models. However, they fail to complete training within the maximum allowed time due to limited resources, such as battery status or CPU power. MFP addresses this issue by reducing resource consumption, thereby enabling the stragglers to participate in FL.
>
> **Clarification on Domain Shifts Example:**
> Images collected by the cameras are not exclusively "multi-view" data, particularly when factors such as lighting conditions or weather vary which are classified as domain shifts. Additionally, images with different view angles only (i.e., "multi-view") are also multi-domain data. For instance, previous work [4] has utilized rotated images to create multi-domain datasets.
>
> **Application Scenarios and Illustration:**
> Thank you for your insightful comments. Due to the length limitations, please refer to **Responses to Weakness 3 \& Question 2** from **Reviewer Ct3S** for the details.
>
> >[1] Lim, Wei Yang Bryan, et al. Federated learning in mobile edge networks: A comprehensive survey. IEEE Commun. Surv. Tutorials 2020.
> >
> >[2] Nguyen, Dinh C., et al. Federated learning for Internet of Things: A comprehensive survey. IEEE Commun. Surv. Tutorials 2021.
> >
> >[3] M. Gerla, et al. Internet of Vehicles: From intelligent grid to autonomous cars and vehicular clouds, WF-IoT 2014.
> >
> >[4] Nguyen, A. Tuan, et al. FedSR: A simple and effective domain generalization method for federated learning. NIPS 2022.
>
>
> ## Response to Weakness 2 \& Question 1 \& Question 2:
>
> **Innovative and Practical Differences of MFP:**
> As detailed in lines 143-160, Algorithm 1, and Fig. 1, a dynamic model fusion mechanism is proposed within MFP. This mechanism is distinct from regular FL. Inspired by transfer learning, DapperFL incorporates the global model into the fine-tuned model to assimilate cross-domain knowledge (formally represented in Eq. 1). Furthermore, a dynamic adjustment mechanism controls the quality of transferred knowledge (as described in Eq. 2), preventing overfitting to the local domain and improving the model's generality.
> Our experiments in the presence of domain shifts show that DapperFL with MFP improves accuracy by up to 7.45\% (on the Digits benchmark) compared to SOTA frameworks that support system heterogeneity.
>
> **Clarification on Additional Computations and Global Model Deployment:**
> It is important to note that MFP does **NOT** entail additional computations. MFP requires clients to fine-tune the model during the initial epoch, consuming resources comparable to regular FL. In subsequent epochs, the pruned model undergoes updates, significantly reducing resource consumption (up to 80\%). Moreover, fine-tuning the global model for at least one epoch is essential in popular FL methods to capture data features, and this is not unique to ours.
>
> **Clarification on Pruning Strategy:**
> We would like to highlight that, the pruning strategy used in DapperFL is structural pruning (channel pruning), rather than unstructural pruning.
> Channel pruning is hardware-friendly and can be easily implemented with popular ML libraries such as PyTorch, making it suitable for edge devices.
> We provide our code in the Abstract, which details the implementation of our approach.
> We want to emphasize that "unstructured pruning" is not presented in our paper. Nevertheless, we acknowledge the importance of clarity and will include a description of the adopted structural pruning.
>
>
> ## Response to Weakness 3:
>
> We appreciate your constructive suggestion. Specifically, the global model transitions to $\mathcal{W}^t$, the fine-tuned model transitions to $\boldsymbol{w}^{t,i}\_{ft}$, and the fused model transitions to $\boldsymbol{w}\_{fs}^{t,i}$.
>
>
> ## Response to Weakness 4:
>
> Thank you for highlighting these studies that offer valuable insights into heterogeneous FL and model pruning. We will incorporate them into our paper to discuss their relationship to ours.
> Despite the similarities, our work contributes novel solutions by focusing on domain shifts and the integration of MFP, DAR, and heterogeneous aggregation.
>
>
> ## Response to Weakness 5:
>
> Thank you for highlighting these important works.
> We did not include distillation-related work as they typically require maintaining both a teacher model and a student model. This imposes a significant burden and is impractical in our targeted scenarios.
> Nonetheless, we acknowledge the relevance of them and will incorporate them to enrich our literature review.
>
>
> ## Response to Limitation 1:
>
> **Hyper-Parameter Selection:**
> Please refer to **Responses to Weakness 1 & Questions 1** from **Reviewer 1ipw** for the details.
>
> **Solution for System Heterogeneity:**
> Please refer to **Response to Weakness 2 \& Question 1 \& Question 2** for the details.
>
> **Theoretical Convergence Analysis:**
> It is noteworthy that extensive prior work has established the convergence of model pruning-based FL [5,6,7,8]. Our DapperFL, which also incorporates model pruning, does not violate the convergence of FL. The experimental results in our paper further support the convergence of DapperFL.
>
> >[5] Zhou, Hanhan, et al. On the convergence of heterogeneous federated learning with arbitrary adaptive online model pruning. arXiv 2022.
> >
> >[6] Li, A., et al. Hermes: an efficient federated learning framework for heterogeneous mobile clients. MobiCom 2021.
> >
> >[7] Zhida Jiang, et al. Fedmp: Federated learning through adaptive model pruning in heterogeneous edge computing. ICDE 2022.
> >
> >[8] Jiang, Yuang, et al. Model pruning enables efficient federated learning on edge devices. TNNLS 2022.

---

> > ### Comment · Reviewer_npsw · 2024-08-09
> > **Response to the rebuttal**
> >
> > Thanks for the authors’ comprehensive rebuttal and the detailed illustrations provided in the pdf file.
> >
> > The responses have effectively addressed my concerns. The clarification on system heterogeneity was particularly helpful. The relevance of DapperFL to the system heterogeneity concern is solved by distinguishing between low-capability clients "failing to complete" training rather than being "unable to perform" it and explaining how MFP mitigates this issue. Additionally, the emphasis on domain shifts and the examples provided have improved the clarity of the scenarios where these issues intersect. My concerns about the suitability of DapperFL is solved by the authors’ clarification on the channel pruning strategy. The authors have mentioned that, DapperFL substantially reduces computation costs in subsequent epochs aside from the minimal and necessary initial costs, which is a significant contribution to the target scenarios. The inclusion of additional references to related work, as well as the reasoning for not focusing on distillation methods, was also satisfactory. Also, I appreciate the decision to include relevant references in the final version. I hope the materials in the rebuttal can be integrated into the final version of the paper to facilitate interested readers’ understanding of the framework.
> >
> > Given the thoroughness and clarity of the responses, I am satisfied that my concerns have been adequately addressed and happy to lift my evaluation of this submission.

---

### Official Review · Reviewer_Ct3S · 2024-07-12

**Soundness:** 3
**Presentation:** 3
**Contribution:** 3
**Rating:** 6
**Confidence:** 4

**Summary:**

The authors propose a new FL scenario that faces challenges of both system heterogeneity and domain shifts. This situation means that the clients have varying capabilities and their data domains also differ. The authors claim that they propose three novel modules to tackle the challenges: Model Fusion Pruning (MFP), Domain Adaptive Regularization (DAR), and a specific aggregation algorithm. The MFP module aims to address the challenge of system heterogeneity. It first fine-tunes the global model with local data to get a local model and then fuses the local model and the global model using a simple weighted average, where the fusion factor decays with the training epochs. Then, it prunes the fused model based on the L1 norm and continues training this model on the local data. The DAR module aims to address the challenge of domain shifts. It adds an L2 regularization on the first few layers of the fused model during the training process. To aggregate the locally pruned models, the models' structure will be discovered using the weights of the global model (of the last communication round) before global aggregation. The authors evaluate their method on two domain generation datasets, and the experimental results show that DapperFL achieves a SOTA performance.

**Strengths:**

1. This paper proposes a novel scenario of Federated Learning (FL) where both the capability and data domains are heterogeneous among clients.
2. Experimental evaluation indicates that the proposed method achieves SOTA performance. This paper also conducts an ablation study to show that each component in the design has a positive effect on the final results.

**Weaknesses:**

1. This paper lacks significant innovation. The proposed framework is just a combination of existing approaches to solving the problems of domain shifts and system heterogeneity, respectively. It’s common to use model pruning to solve the problem of system heterogeneity, such as [1][2][3]. The model aggregation method seems natural for updating the subnetwork of the local model on the global model and has already been proposed in similar work such as [3].
2. The explanation of the proposed method is unclear. For example, the Domain Adaptive Regularization module does not specify how the model is segmented.
3. The motivation of this paper is not strong enough. This paper lacks evidence to demonstrate the importance of the scenarios present in the Introduction section (with both system heterogeneity and domain heterogeneity).

[1] Jiang, Z., Xu, Y., Xu, H., Wang, Z., Liu, J., Chen, Q., & Qiao, C. (2023). Computation and communication efficient federated learning with adaptive model pruning. *IEEE Transactions on Mobile Computing*, *23*(3), 2003-2021.
[2] Li, A., Sun, J., Zeng, X., Zhang, M., Li, H., & Chen, Y. (2021, November). Fedmask: Joint computation and communication-efficient personalized federated learning via heterogeneous masking. In *Proceedings of the 19th ACM Conference on Embedded Networked Sensor Systems* (pp. 42-55).
[3] Li, A., Sun, J., Li, P., Pu, Y., Li, H., & Chen, Y. (2021, October). Hermes: an efficient federated learning framework for heterogeneous mobile clients. In *Proceedings of the 27th Annual International Conference on Mobile Computing and Networking* (pp. 420-437).

**Questions:**

1. How do you segment the model in the Domain Adaptive Regularization module?
2. Do you have any evidence showing the importance or prevalence of the scenario present in the Introduction section (FL with both system and domain heterogeneity)?

**Limitations:**

The authors discuss the limitation that the proposed method introduces four hyper-parameters and the choice of these hyperparameters significantly impacts the performance of the method.

---

> ### Author Rebuttal · Authors · 2024-08-06
>
> We sincerely thank you for your constructive and helpful comments. Below we address your concerns in order.
>
>
> ## Response to Weakness 1:
>
> We would like to emphasize that, our proposed framework is **NOT** a combination of existing approaches.
> While our work builds on established concepts, it introduces novel components and methods that address significant gaps in the current literature, particularly concerning domain shifts in resource-constrained edge environments.
> We outline the substantial differences and innovative contributions of our DapperFL framework from existing works as follows:
>
> 1) **Transfer Learning-Inspired Model Fusion Pruning.**
> Existing heterogeneous FL frameworks, such as [1,2,3], focus on addressing data heterogeneity (e.g., variations in sample and label distributions) but do not directly tackle the challenge of domain shifts, which can significantly impact the performance of pruned models.
> Our work introduces Model Fusion Pruning (MFP), which not only accounts for data heterogeneity but also addresses domain shifts. MFP dynamically combines local domain information with cross-domain information to prune local models effectively, enhancing their performance under domain shift conditions.
>
> 2) **Lightweight Solutions for Domain Shifts.**
> Current methods often require extensive data collection transmission to handle domain shifts, which is impractical in resource-constrained environments.
> Our DapperFL framework, however, offers a novel resource-efficient approach to handling domain shifts through the MFP and Domain Adaptive Regularization (DAR) modules. The MFP module prunes local models leveraging local and cross-domain information. The DAR module introduces a regularization term that encourages the generation of domain-invariant representations at the local level, eliminating the need for transmitting domain-invariant representations, which is commonly required in existing studies.
>
> 3) **Specific Model Aggregation in Cross-Domain Environments.**
> Existing works, including [3], lack mechanisms to ensure that domain-specific features are retained in heterogeneous environments.
> In contrast, we propose a novel model recovery mechanism during heterogeneous model aggregation. This mechanism ensures the preservation of specific domain knowledge while transferring global knowledge, as detailed in Eq.6 of our original paper. This unique operation reinstates the pruned model's architecture and incorporates global knowledge, differentiating our approach from existing methods.
>
> >[1] Jiang, Z., et al. Computation and communication efficient federated learning with adaptive model pruning. TMC, 2023.
> >
> >[2] Li, A., et al. Fedmask: Joint computation and communication-efficient personalized federated learning via heterogeneous masking. SenSys 2021.
> >
> >[3] Li, A., et al. Hermes: an efficient federated learning framework for heterogeneous mobile clients. MobiCom 2021.
>
>
> ## Response to Weakness 2 \& Question 1:
>
> We follow a conventional setting commonly used in representation learning research to segment the model. Specifically, all layers except the final linear layer act as the encoder, while the last linear layer of the model acts as the predictor.
> We acknowledge the importance of clarity and will include a detailed description of this segmentation process.
>
>
> ## Response to Weakness 3 \& Question 2:
>
> Consider a mobile voice assistant like Apple Siri, which requires an efficient FL framework to collect and analyze user voice data [4]. Users may interact with Siri using different accents even when speaking the same language. This variation leads to domain shift problems, as the system must adapt to different speech patterns. Additionally, the diversity in mobile phone capabilities introduces system heterogeneity, as devices with varying processing powers may affect the performance of local computations. Another pertinent example is the IoV network, where smart vehicles collect, compute and communicate data for tasks like navigation and traffic management [3,5]. The data collected, such as images, can vary significantly in quality due to factors like motion blur, leading to multi-domain datasets. On the other hand, the varying computing power and network conditions of vehicles create system heterogeneity challenges [5]. Beyond these scenarios, our framework is relevant to various scenarios such as smart homes and smart health systems [1,2].
>
> We have added Fig. 1 to the rebuttal PDF to visually demonstrate the impact of these issues on FL performance. Specifically:
> 1) **System Heterogeneity:** Device 1, with stringent resource constraints, fails to complete its model update within the maximum allowed time. As a result, the central server excludes it from the FL process and misses out on the data features captured by Device 1.
> 2) **Domain Shifts:** Devices 2 and 3 collect data from different domains, leading to diverse local models. With non-IID data, the global model aggregation may become biased, particularly if Device 3 contributes more data, thus potentially limiting the benefit for Device 2.
>
> >[1] Lim, Wei Yang Bryan, et al. Federated learning in mobile edge networks: A comprehensive survey. IEEE Commun. Surv. Tutorials 2020.
> >
> >[2] Nguyen, Dinh C., et al. Federated learning for Internet of Things: A comprehensive survey. IEEE Commun. Surv. Tutorials 2021.
> >
> >[3] M. Gerla, et al. Internet of Vehicles: From intelligent grid to autonomous cars and vehicular clouds, WF-IoT 2014.
> >
> >[4] Paulik, Matthias, et al. Federated evaluation and tuning for on-device personalization: System design \& applications. arXiv 2021.
> >
> >[5] D. Ye, et al. Federated learning in vehicular edge computing: A selective model aggregation approach, IEEE Access 2020.
>
>
> ## Response to Limitation 1:
>
> Thank you for your insightful comments. Due to the rebuttal's length limitations, please refer to our **Responses to Weakness 1 \& Questions 1** from **Reviewer 1ipw** for detailed information.

---

> > ### Comment · Reviewer_Ct3S · 2024-08-11
> >
> > Thanks for your detailed rebuttal response. In the rebuttal, the authors have elaborated on the novelty of this paper, and I think their claims are reasonable. Besides, the authors have clarified the previously unclear description of the methodology, i.e. the model's segmentation approach. Finally, the authors have provided specific and meaningful examples, demonstrating that the scenarios in the paper are common and significant. Therefore, our issues have been well addressed and we are satisfied with the rebuttal response. We will raise our minor scores to 3/3/3 points and the overall rating to 6 (Weak Accept).

---

### Official Review · Reviewer_pq1G · 2024-07-12

**Soundness:** 3
**Presentation:** 4
**Contribution:** 4
**Rating:** 7
**Confidence:** 5

**Summary:**

This paper proposes DapperFL, an innovative FL framework designed to enhance model performance across multiple domains within heterogeneous FL environments. DapperFL addresses the system heterogeneity challenge through the deployment of a dedicated Model Fusion Pruning (MFP) module. Additionally, a Domain Adaptive Regularization (DAR) module is introduced to improve the overall performance of DapperFL. The evaluation results are positive compared with SOTA.

**Strengths:**

S1: The proposed framework is novel in FL to apply local-knowledge-based heterogenous pruning to the problem of domain shift in FL.
S2: This paper achieves better model performance with fewer resource consumptions on a real-world FL platform, which is a significant technical contribution to FL.
S3: The paper is easy to follow, with a clear presentation and publicly available source code.

**Weaknesses:**

W1: It could be better to provide a discussion on why only one epoch is enough to fine-tune the global model in Algorithm 1.
W2:	In the evaluation section, the authors should provide more framework-specific default values for hyper-parameters.
W3: The subscript “i” is used to denote both client subscripts in the methods section and client-level subscripts in the experiments section. It would be better to use different notation.
W4: The limitations of this work could be elaborated more to let the audience develop a comprehensive understanding.

**Questions:**

Q1: Why is the l_1 norm (rather than other normalization methods such as the l_2 norm) used to calculate mask matrices?
Q2: When DapperFL aggregates heterogenous models on the central server, how does the sever know the architecture of these heterogenous models?

**Limitations:**

See weaknesses.

---

> ### Author Rebuttal · Authors · 2024-08-06
>
> We sincerely thank you for your constructive and helpful comments. Below we address your concerns in order.
>
> ## Response to Weakness 1:
> We utilize one epoch of local training to determine the pruned models for the following reasons:
>
> 1) Additional local epochs do not significantly enhance the model's performance, which justifies the use of a single epoch for efficiency. As noted in [1], experiments have demonstrated that extending local training beyond one epoch yields results comparable to those achieved with just one epoch.
>
> 2) In previous domain generalization-related FL research, such as [1], one epoch is also employed to collect local domain information. This method has proven adequate for capturing essential features and domain characteristics.
>
> 3) Pioneering research in model design and neural architecture search, such as in [2], has demonstrated that a few epochs are sufficient to obtain a coarse estimate of a sub-model. This approach is effective in quickly assessing model configurations without extensive computation.
>
>
> >[1] Zhang, Jianqing, et al. Eliminating domain bias for federated learning in representation space. NIPS 2024.
> >
> >[2] Tan, Mingxing, et al. Mnasnet: Platform-aware neural architecture search for mobile. CVPR 2019.
>
> ## Response to Weakness 2:
> We have provided the default values for three framework-specific hyper-parameters in FedProx and MOON, as well as all four hyper-parameters in our DapperFL framework. Additionally, we have included more default values for the hyper-parameters in the comparison frameworks in the rebuttal PDF file, which will be incorporated into our final version. Specifically, the default values are $\alpha^{L2R}=0.01$ in FedSR, $\alpha^{CMI}=0.001$ in FedSR, and $\tau=0.02$ in FPL. These values are consistent with those used in the original implementations of these frameworks, ensuring accuracy and reliability in our comparisons.
>
> ## Response to Weakness 3:
> Thank you for your meticulous observation. To avoid confusion and ensure clarity, we will change the notation for client-level subscripts in the experiments section to $l$. We hope this adjustment will improve the readability and precision of our paper.
>
> ## Response to Weakness 4:
> We acknowledge that further elaboration on the limitations of this work could provide the reader with a fuller understanding. As an example, the limitations of hyper-parameter selection will be described below:
>
> “Despite DapperFL shows promise in addressing system heterogeneity and domain shifts, it also introduces four hyper-parameters: $\alpha_0$, $\alpha_{min}$, $\epsilon$, and $\gamma$. These hyper-parameters influence the domain generalization performance of the global model. One limitation is the manual tuning required for these hyper-parameters, which may be time-consuming and require domain-specific expertise. As such, a potential future direction is the development of an automatic selection mechanism for these hyper-parameters. This enhancement would not only improve the flexibility of DapperFL but also make it more accessible to users without extensive knowledge in hyper-parameter tuning, thereby broadening the applicability and ease of deployment of our framework.”
>
> ## Response to Question 1:
> We opted for the $\ell_1$ norm instead of the $\ell_2$ norm for the following reasons: The $\ell_1$ norm requires fewer computational resources, making it more suitable for our framework, which targets resource-constrained edge devices.
> Additionally, as shown by pioneering work [3], there is no significant difference in the effectiveness of using $\ell_1$ norm versus $\ell_2$ norm for calculating mask matrices in the context of model pruning. This finding supports our choice, as the $\ell_1$ norm offers a more resource-efficient alternative without compromising the quality of the pruning process.
>
> >[3] Li, Hao, et al. Pruning Filters for Efficient ConvNets. ICLR 2022.
>
> ## Response to Question 2:
> In our proposed framework, the heterogeneous models are transmitted to the server along with the mask matrices. These mask matrices indicate the architecture of the heterogeneous models.
> Specifically, the mask matrices encode the structure of each model by highlighting which parts of the model are retained and which are pruned.
> This information allows the central server to understand and manage the varying architectures of the heterogeneous models during aggregation.

---

> > ### Comment · Reviewer_pq1G · 2024-08-12
> >
> > Thanks for the detailed responses. I'm generally fine with the results.  Thus, I will keep my score.

---

### Official Review · Reviewer_1ipw · 2024-07-27

**Soundness:** 4
**Presentation:** 4
**Contribution:** 3
**Rating:** 6
**Confidence:** 5

**Summary:**

The paper introduces a novel federated learning framework to address system heterogeneity and domain shifts in edge computing environments. The framework employs a Model Fusion Pruning (MFP) module to generate personalized compact local models and a Domain Adaptive Regularization (DAR) module to enhance performance across multiple domains. Experimental results demonstrate the effectiveness of the proposed method in terms of accuracy improvements and model compression.

**Strengths:**

1. The proposed method is an innovative approach in edge environments. The framework’s ability to simultaneously tackle heterogeneous and low-resource challenges in edge environments. The proposed solution is highly applicable and effective in real-world edge computing scenarios.

2. The paper’s contents are well-organized. The motivation is clearly illustrated and well supported by the proposed method.

3. The experimental analysis is comprehensive.  For example, the chosen benchmark datasets include MNIST, USPS, SVHN, SYN, Caltech, Amazon, Webcam, and DSLR.  The comparisons with 8 SOTA frameworks add credibility to the claims.

4. The proposed method is technique sounds. Moreover, the source codes are provided to ensure the reproducibility of this work.

**Weaknesses:**

1. The proposed method is sensitive to hyperparameter selection. For example, the framework introduces several hyper-parameters (α0, αmin, ϵ, γ) that significantly influence performance, which also complicates the application of DapperFL. As pointed out in the “Limitations” of this manuscript, it could benefit from providing a set of default values for these hyper-parameters or an automatic hyper-parameter selection mechanism.

2. Experimental Results Analysis: Despite DapperFL achieving competitive model accuracy with a significantly smaller model footprint, it would be beneficial to discuss why a smaller model could outperform a larger one.

3. It is unclear how the proposed meethod to be applied to new devices in test or production phase.

4. Missing some related work, such as heterogenous FL [1,2] and model pruning on devices [3,4].

[1] Zhuangdi Zhu, et al., Data-free knowledge distillation for heterogeneous federated learning. In International conference on machine learning (pp. 12878-12889). ICML 2021.

[2] Yue Tan, et al., Federated Prototype Learning across Heterogeneous Clients, AAAI 2022

[3] Yuang Jiang, et al., Model Pruning Enables Efficient Federated Learning on Edge Devices, IEEE TNNLS 2022

[4] Haiyan Zhao, et al., One-Shot Pruning for Fast-adapting Pre-trained Models on Devices, arXiv preprint arXiv:2307.04365

**Questions:**

1. Any suggestions when choosing hyperparameters (α0, αmin, ϵ, γ) for good global model performance in DapperFL?

2. Why does the smaller model generated by DapperFL have an accuracy comparable to that of SOTA, and sometimes even exceed the accuracy of a full-size model?

3.  In eq.6, the authors combine “local knowledge” and “global knowledge” to recover the local model. It could be better to introduce these definitions more clearly before using them. What do “local knowledge” and “global knowledge” in DapperFL stand for, and where are they derived from?

---

> ### Author Rebuttal · Authors · 2024-08-06
>
> We sincerely thank you for your constructive and helpful comments. Below we address your concerns in order.
>
> ## Response to Weakness 1 \& Question 1:
> Thanks for your suggestion to provide default values or an automatic selection mechanism for hyper-parameters. We acknowledge that hyper-parameter selection is indeed crucial, especially for practitioners looking to effectively apply the DapperFL framework.
>
> In DapperFL, $\alpha \in [\alpha_{min}, \alpha_0]$ control the balance between local and global information, which can impact how well the model generalizes across different domains. The sensitivity factor $\epsilon$ determines the rate at which $\alpha$ decreases from $\alpha_0$ to $\alpha_{min}$, while $\gamma$ influences the regularization strength.
> In our paper, we have conducted ablation experiments (in Section 4.3) to identify suitable hyper-parameter settings. The experimental results suggest that:
> 1) Increasing $\alpha_0$ or decreasing $\alpha_{min}$ generally enhances the incorporation of global knowledge, improving generalization. In our experiments, setting $\alpha_0$ to 0.9 and $\alpha_{min}$ to 0.1 resulted in good model accuracy.
> 2) As $\epsilon$ increases, model accuracy suffers from a decrease due to insufficient guidance from the global knowledge except for $\epsilon$ is less than 0.2.
> 3) $\gamma$ helps regularize the model and prevent overfitting. In our experiments, setting it to an optimal value of 0.01 can balance model personalization on the local domain and generalization across multiple domains.
>
> Furthermore, recognizing the abnormal relationship between model accuracy and hyper-parameter $\epsilon$ as it is less than 0.2, we implement Bayesian search as an automatic selection mechanism to find a better $\epsilon$. Due to limited rebuttal time and hardware constraints, we have tried our best to include as many single runs as possible to search for a better $\epsilon$ for DapperFL. Specifically, we run DapperFL on the Office Caltech benchmark 40 times, adopting a distinct $\epsilon$ of less than 0.2 each time. The values are selected using the Bayesian search algorithm. The results are presented in Fig. 2 of the rebuttal PDF file. As illustrated, the Bayesian search-based automatic selection mechanism indicates that model accuracy is likely to reach a higher level when $\epsilon$ approaches 0.2, aligning with our original paper's default setting of $\epsilon=0.2$.
>
>
> ## Response to Weakness 2 \& Question 2:
> The relationship between model accuracy and model footprint is complex and not necessarily linear, as evidenced by previous studies [1, 2]. In our work, we employ the MFP module within DapperFL, which addresses the over-fitting issues that can arise from the over-parameterization of neural networks. This module not only reduces the model footprint but also enhances the model's generalization capabilities, leading to more robust performance across diverse domains. Additionally, the DAR module is incorporated to further improve model accuracy, particularly in scenarios involving distributed clients with domain shifts.
>
> > [1] Z. Liu, et al. Learning efficient convolutional networks through network slimming, ICCV 2017.
> >
> > [2] S. Shen, et al. Efficient deep structure learning for resource-limited IoT devices, GLOBECOM 2020.
>
>
> ## Response to Weakness 3:
> One primary objective of DapperFL is to optimize a robust global model with strong domain generalization capabilities across distributed local data. In the test or production phase, this optimized global model can be deployed on new devices, ensuring good performance across different environments. Additionally, newly introduced devices can also participate in the DapperFL framework. They can contribute to the continual improvement of the global model by optimizing a local model based on their own local data, which is then aggregated to update the global model. This approach not only enhances the global model's adaptability to new data distributions but also leverages the unique data available on new devices to improve overall model robustness.
>
>
> ## Response to Weakness 4:
> Thank you for highlighting these relevant studies. These studies offer valuable insights into heterogeneous FL and model pruning on devices, aligning well with the themes of our research. We will incorporate these references into our paper to discuss their relationship to our study, highlighting similarities, differences, and how our work extends or diverges from these existing approaches.
>
>
> ## Response to Question 3:
> In our work, “local knowledge” refers to the feature extraction capabilities of the local model, which are derived from the specific data available in its local domain. This knowledge encapsulates the nuances and characteristics of the data that the local model has been trained on.
> On the other hand, “global knowledge” represents the aggregated feature extraction capabilities of the global model, which are informed by data across all participating domains. The global model synthesizes diverse knowledge from different local models to provide a more generalized understanding that is applicable across multiple domains. By combining these two forms of knowledge, we aim to leverage both the specialized insights of local models and the generalized capabilities of the global model, thereby enhancing the performance and adaptability of DapperFL.

---

### Author Rebuttal · Authors · 2024-08-06

We sincerely thank the reviewers for their time and appreciate all the detailed reviews and constructive feedback. Extra experimental results and illustrations are presented in the rebuttal PDF file to address some common concerns raised by reviewers. In addition, each review will be replied to individually. The discussion here will be properly incorporated into a new version of our paper.

---

### Decision · Program_Chairs · 2024-09-25

**Decision:**

Accept (oral)

**Comment:**

The paper proposes a novel application scenario. The paper is well written. The unclear parts are resolved by rebuttal. All reviewers agreed to accept this paper.